# Thiobarbiturate-Derived Compound MHY1025 Alleviates Renal Fibrosis by Modulating Oxidative Stress, Epithelial Inflammation, and Fibroblast Activation

**DOI:** 10.3390/antiox12111947

**Published:** 2023-10-31

**Authors:** Jeongwon Kim, Jieun Lee, Dahye Yoon, Minjung Son, Mi-Jeong Kim, Sugyeong Ha, Doyeon Kim, Ji-an Yoo, Donghwan Kim, Hae Young Chung, Hyung Ryong Moon, Ki Wung Chung

**Affiliations:** 1Department of Pharmacy, Research Institute for Drug Development, College of Pharmacy, Pusan National University, Busan 46241, Republic of Korea; 98juon_k@naver.com (J.K.); min30124@naver.com (M.S.); yos3552@naver.com (M.-J.K.); tnrn34@hanmail.net (S.H.); kdy991117@naver.com (D.K.); choiceyja@naver.com (J.-a.Y.); hyjung@pusan.ac.kr (H.Y.C.); 2Department of Manufacturing Pharmacy, Research Institute for Drug Development, College of Pharmacy, Pusan National University, Busan 46241, Republic of Korea; yijiun@pusan.ac.kr (J.L.); dahae0528@pusan.ac.kr (D.Y.); 3Functional Food Materials Research Group, Korea Food Research Institute, Wanju-gun 55365, Republic of Korea; kimd@kfri.re.kr

**Keywords:** kidney fibrosis, oxidative stress, inflammation, fibroblast, NF-κB

## Abstract

Chronic kidney disease (CKD) is a kidney structure and function abnormality. CKD development and progression are strongly influenced by oxidative stress and inflammatory responses, which can lead to tubulointerstitial fibrosis. Unfortunately, there are no effective or specific treatments for CKD. We investigated the potential of the thiobarbiturate-derived compound MHY1025 to alleviate CKD by reducing oxidative stress and inflammatory responses. In vitro experiments using NRK52E renal tubular epithelial cells revealed that MHY1025 significantly reduced LPS-induced oxidative stress and inhibited the activation of the NF-κB pathway, which is involved in inflammatory responses. Furthermore, treatment with MHY1025 significantly suppressed the expression of fibrosis-related genes and proteins induced by TGFβ in NRK49F fibroblasts. Furthermore, we analyzed the MHY1025 effects in vivo. To induce kidney fibrosis, mice were administered 250 mg/kg folic acid (FA) and orally treated with MHY1025 (0.5 mg/kg/day) for one week. MHY1025 effectively decreased the FA-induced inflammatory response in the kidneys. The group treated with MHY1025 exhibited a significant reduction in cytokine and chemokine expression and decreased immune cell marker expression. Decreased inflammatory response was associated with decreased tubulointerstitial fibrosis. Overall, MHY1025 alleviated renal fibrosis by directly modulating renal epithelial inflammation and fibroblast activation, suggesting that MHY1025 has the potential to be a therapeutic agent for CKD.

## 1. Introduction

Chronic kidney disease (CKD) is a prevalent condition that affects approximately 10% of the global population, and its limited treatment options contribute to high mortality rates [1]. CKD frequently progresses to end-stage renal disease, a life-threatening condition that necessitates renal replacement therapy such as dialysis or kidney transplantation. The development of kidney fibrosis is a key pathological process in CKD. To develop therapies that can prevent or slow the progression of CKD, it is crucial to understand the mechanisms underlying renal fibrosis [2]. Fibrosis refers to the excessive accumulation of extracellular matrix (ECM) primarily by fibroblasts residing in the kidney tissue [3,4]. Normally, the kidneys maintain a minimal amount of ECM to support their structure and function. Upon tissue injury, wound-healing mechanisms are activated to limit inflammation and promote proper tissue regeneration. However, persistent inflammatory responses impede complete regeneration, resulting in fibrotic scar tissue formation. Excessive deposition of ECM during chronic and pathological fibrosis disrupts the normal architecture of the kidney and impairs its function [5]. Eventually, unresolved kidney fibrosis reaches an irreversible stage and contributes to renal failure.

Extensive research has focused on unraveling the mechanisms involved in the development of kidney fibrosis. Regardless of the causative factors, excessive inflammatory responses are observed in this process. Renal fibrosis is triggered and augmented by the inflammatory response. Multiple cell types, including myeloid-derived inflammatory cells, fibroblasts, pericytes, epithelial cells, and endothelial cells, participate in inflammation [6]. Recently, an increasing number of studies have focused on the role of tubular epithelial cells (TECs) in renal fibrosis and inflammation [7,8,9]. Once damaged, TECs secrete major cytokines and chemokines that play important roles in fibrosis and inflammation [7]. TECs directly secrete pro-fibrotic cytokines, including transforming growth factor-β (TGFβ) and connective tissue growth factor, thus activating nearby fibroblasts to produce ECM proteins. Furthermore, TECs secrete major chemokines, including CCL2 and CCL5, that recruit myeloid-derived inflammatory cells [9]. However, the role of infiltrated myeloid-derived inflammatory cells in fibrosis remains unknown. Interestingly, several recent studies have indicated that bone marrow-derived cells directly contribute to collagen production in kidney fibrosis models [10].

Nuclear factor kappa-light-chain-enhancer of activated B cells (NF-κB) is a crucial transcription factor involved in the regulation of inflammation. It is present in nearly all types of cells and tissues and responds to various internal and external stimuli. These stimuli include infections, inflammatory cytokines, oxidative stress, and chemical agents [11]. NF-κB plays a vital role in maintaining normal immune responses against infections; however, when its activation becomes dysregulated, it becomes a significant contributor to inflammatory diseases. NF-κB activation has commonly been observed during kidney injury, whether caused by infection or occurring in a sterile environment [12,13]. In models of ischemic acute kidney injury, NF-κB activity has been detected in tubular epithelial cells [14]. The activation of NF-κB in this context worsens tubular injury and intensifies an inappropriate inflammatory response. Additionally, angiotensin activates NF-κB in tubular epithelial cells, leading to an increased inflammatory response [15]. Conversely, inhibiting NF-κB activity protects against renal inflammation [16]. Therefore, targeting NF-κB specifically in tubular epithelial cells represents an interesting approach to the regulation of kidney diseases.

Previous studies have shown that thio-barbiturate-derived compounds have antioxidant effects and prevent LPS-induced inflammation in the liver [17]. Thio-barbiturate-derived compounds showed significant ROS- and ONOO-scavenging effects in test tubes [17]. In this study, the authors showed that one of the thio-barbiturate-derived compounds, MHY1025, significantly reduced LPS-induced oxidative stress and NF-κB activation in both macrophage and liver injury models. Moreover, it has been reported by others that barbiturate compounds exhibit antioxidant properties [18]. However, the effects of these compounds on general aspects of renal fibrosis have not been investigated. Here, we evaluated the antioxidative and anti-inflammatory effects of MHY1025 in renal epithelial cells in vitro. Furthermore, we demonstrated the antifibrotic and anti-inflammatory effects of MHY1025 in mouse models of folic acid-induced renal fibrosis. MHY1025 treatment significantly reduced fibrosis and inflammation in a mouse model of renal fibrosis.

## 2. Materials and Methods

### 2.1. Animal Studies

All animal experiments were performed in accordance with the guidelines for animal experimentation issued by Pusan National University (PNU) and were approved by the Institutional Animal Care Committee of PNU (IACUC approval No. PNU-2022-3180). The C57BL/6J mice were purchased from Hyochang Science (Daegu, Korea). All mice were maintained at 23 ± 2 °C with a relative humidity of 60 ± 5% and 12 h light/dark cycles and were provided with free access to water and food. The animals were randomly sorted into four different groups (*n* = 7~8). To examine the therapeutic effect of MHY1025 in a renal fibrosis model, male mice were intraperitoneally administered 250 mg/kg of folic acid dissolved in 0.3 M NaHCO_3_ or a vehicle. Throughout the experimental period, a daily oral gavage was administered for one week, delivering either 0.5 mg/kg MHY1025 dissolved in 1% DMSO in water or a solution of 1% DMSO alone. Following the sacrifice of the mice, the kidneys and sera were collected. The kidneys were promptly preserved at −80 °C or fixed in 10% neutral formalin for histochemical analysis. Serum samples were collected by centrifugation at 3000 rpm for 20 min at 4 °C. Blood urea nitrogen (BUN) levels were measured using a commercial assay kit from Shinyang Diagnostics (SICDIA L-BUN, 1120171, Seoul, Korea) according to the manufacturer’s instructions.

### 2.2. Cell Culture Experiments

The rat tubular epithelial cell line (NRK52E) and rat kidney fibroblasts (NRK49F) were purchased from the ATCC. All cell lines were cultured in an incubator at 37 °C in humidified 95% air and 5% CO_2_ in Dulbecco’s modified Eagle’s medium (DMEM) containing 5% fetal bovine serum (FBS) and 1% penicillin. A stock solution of MHY1025 was prepared at a concentration of 10 mM in DMSO and stored at −20 °C until use. To determine the antioxidant and anti-inflammatory effects of MHY1025, LPS (L2630, Sigma, St. Louis, MO, USA) treatment was used in NRK52E cells. To evaluate the inhibitory effect of MHY1025 on fibrotic cell activation, NRK49F cells were pretreated with MHY1025 and then treated with 10 ng/mL of TGF-β1 (100-21, PeproTech, Cranbury, NJ, USA) for 24 h. The concentration range of MHY1025 administered to the cells ranged from 1 to 20 μM, as determined by the experimental conditions. All the cell culture experiments were repeated at least 3 times.

### 2.3. Cell Viability Assay

Cell viability was measured to determine the cytotoxicity of MHY1025 in NRK52E cells. The EZ-cytox Cell Viability Assay Kit (Dogen, Seoul, South Korea) was used according to the manufacturer’s instructions. Briefly, NRK52E cells were cultured to 70–80% confluence in serum-free media and treated with MHY1025 at the indicated concentrations for 24 h. The cells were further cultured for 30 min by adding the cell viability solution in the incubator, and the absorbance was measured at 450 nm using a spectrophotometer. The percentage of viable cells was calculated according to the following formula: [(chemically treated group/control group) × 100].

### 2.4. Determination of Antioxidant Effect

Cellular oxidative stress was quantified to determine the antioxidant effects of MHY1025. ROS generation was measured using a 2′,7′-dichlorodihydrofluorescein diacetate (DCFDA) fluorescent dye. Briefly, the cells were scarpered and centrifuged and the supernatants were removed. Cells were incubated with 50 μM DCFDA dissolved in PBS. Changes in fluorescence were detected using a GENios plate reader (Tecan Instruments, Salzburg, Austria) at excitation and emission wavelengths of 485 and 530 nm, respectively. Changes in the ROS levels were detected using a fluorescence microscope. The cellular glutathione (GSH) levels were quantified. Briefly, 25% of the meta-phosphoric acid-added cell pellets were centrifuged at 12,000 rpm for 10 min and the supernatant was collected for the assay. For GSH, 1 mM EDTA-50 mM phosphate buffer was added to the supernatant, followed by the addition of o-phthaladehyde. After 20 min at room temperature, the fluorescence was measured at excitation and emission wavelengths of 360 and 460 nm, respectively.

### 2.5. Determination of the Transcriptional Activity

A luciferase assay was performed to detect the transcriptional activity of NF-κB in the NRK52E cells. Lipofectamine 3000 transfection reagent (Invitrogen, Carlsbad, CA, USA) and Opti-MEM (Gibco, Grand Island, NY, USA) were used according to the manufacturer’s instructions for the plasmid transfection. Cells were cultured to 70–80% confluence in serum-free media and transfected with 0.05 ng of NF-κB promoter-LUC plasmids. After 24 h of NF-κB transfection, the cells were treated with 10 μM MHY1025 for 2 h. Luciferase activity was measured using the One-Glo Luciferase Assay System (Promega, Madison, WI, USA) and detected using a luminescence plate reader (Berthold Technologies GmbH & Co., Bad Wildbad, Germany).

### 2.6. Immunofluorescence

Immunofluorescence staining (IF) was used to detect the location of proteins in the cells. Briefly, cells were fixed in 4% formaldehyde for 10 min, washed three times with ice-cold phosphate-buffered saline (PBS), and exposed to 0.25% Triton-X 100 in PBS for 10 min for permeabilization. After blocking with 1% BSA/0.1% Tween 20 in PBS at room temperature for 30 min, the cells were incubated overnight with primary antibody diluted in blocking buffer at 4 °C. The cells were washed with PBS, incubated with the secondary antibody for 1 h in the dark, and counterstained with Hoechst 33,258 in PBS for 1 min. The images were acquired using a fluorescence microscope (LS30; Leam Solution, Seoul, Korea).

### 2.7. Protein Extraction and Western Blot Analysis

ProEXTM CETi (Translab, Daejeon, Korea) containing a protease inhibitor cocktail and phosphate inhibitor was used to extract total protein lysates from the tissues. Total protein was extracted from cells using RIPA buffer (#9806, Cell Signaling Technology, Danvers, MA, USA) containing a protease inhibitor cocktail. Throughout the extraction process, all tubes, solutions, and centrifuges were maintained at a temperature of 4 °C. To determine protein concentrations, BCA reagent from Thermo Scientific (Waltham, MA, USA) was employed. Extracted proteins (10–20 μg) underwent a 5 min boiling step with 4× sample buffer (Cat#1610747, Bio-Rad, CA, USA) in a volume ratio of 3:1. Following protein extraction, sodium dodecyl sulfate-polyacrylamide gel electrophoresis was used to separate proteins from each sample on 8–15% acrylamide gels. Subsequently, a Bio-Rad Western blot system was used to transfer proteins to polyvinylidene difluoride (PVDF) membranes (Millipore, Burlington, MA, USA). The membranes were promptly placed in blocking buffer (5% non-fat milk) containing 10 mM Tris (pH 7.5), 100 mM NaCl, and 0.1% Tween 20. The membranes were subjected to a 30 min wash with Tris-buffered saline (TBS; 50 mM Tris and 150 mM NaCl [pH 7.6]) and Tween buffer (10 mM Tris [pH 7.5], 100 mM NaCl, and 0.1% Tween 20). The membranes were incubated with specific primary antibodies (diluted at 1:500 to 1:2000; Appendix A) at a temperature of 4 °C overnight. After the primary antibody incubation, the membranes were washed three times with TBS-Tween buffer for 10 min each, followed by incubation with a horseradish peroxidase-conjugated anti-mouse, anti-rabbit, or anti-goat secondary antibody (1:10,000 dilution) at 25 °C for 1 h. Immunoblots were visualized using Western Bright Peroxide solution from Advansta (San Jose, CA, USA) and a ChemiDoc imaging system from Bio-Rad, following the manufacturer’s instructions. Western blot analysis was performed on the kidneys of every animal tested in the experiments, and the figures display representative results.

### 2.8. RNA Extraction and qPCR

Total RNA was extracted using the TRIzol reagent (Invitrogen, Carlsbad, CA, USA) according to the manufacturer’s guidelines. Briefly, kidney tissues or cells were homogenized in the TRIzol reagent. Subsequently, 0.2 mL of chloroform per 1 mL of homogenate was added, and the samples were vigorously shaken for 15 min. The resulting aqueous phases were transferred to fresh tubes and an equal volume of isopropanol was added. Following a 15 min incubation at 4 °C, the samples were centrifuged at 12,000× *g* for 15 min at 4 °C. The supernatants were carefully decanted and the RNA pellets were washed with 75% ethanol by vortexing. Afterward, centrifugation at 7500× *g* for 8 min at 4 °C was performed. The resulting pellets were air-dried for 10–15 min and dissolved in diethyl pyrocarbonate-treated water. For reverse transcription, 1.0 μg of isolated RNA was used with a cDNA synthesis kit from GenDEPOT (Katy, TX, USA). Quantitative real-time polymerase chain reaction (qPCR) was performed using SYBR Green Master Mix (BIOLINE, Taunton, MA, USA) and the CFX Connect System (Bio-Rad). The primers were designed using Primer3Plus (Appendix A). To analyze qPCR data, the 2-ΔΔCT method was employed as a relative quantification approach.

### 2.9. Histological Analysis

To examine histological alterations in the kidneys, kidney specimens were fixed in a 10% neutral formalin solution. Paraffin-embedded sections were prepared and stained with hematoxylin and eosin (H&E) staining. To assess the extent of renal fibrosis and injury, Sirius Red (SR) staining was performed using a commercially available kit (VB-3017; Rockville, MD, USA). Immunohistochemical analysis was performed to evaluate protein expression in the kidneys. Paraffin-embedded sections were incubated with specific primary antibodies as indicated. Visualization was achieved using diaminobenzidine substrates and counterstaining was performed using hematoxylin. Images were captured using a microscope (LS30).

### 2.10. In Situ Hybridization

In situ hybridization (ISH) was performed on tissue samples that had been fixed in formalin and embedded in paraffin. For visualization of RNA expression within the tissue, either the RNAscope 2.5 HD Assay (322300, Biotechne, Minneapolis, MN, USA) or the RNAscope 2.5 HD Duplex Detection Kit (322436, bio-techne, Minneapolis, MN, USA) was used, following the manufacturer’s instructions. The RNAscope assay was conducted using the following probes: Mm-Ccl2 cat# 311791, Mm-Emr1 cat# 317969-C2, and Mm-Col1a1 cat# 319379. A microscope (LS30) was used to capture images.

### 2.11. Quantification and Statistical Analysis

Student’s *t*-test was used to analyze differences between two groups, and analysis of variance was used to analyze intergroup differences. Statistical significance was set at *p* < 0.05. Statistical analyses were performed using GraphPad Prism version 5 (GraphPad Software Inc., San Diego, CA, USA). Image calculations were performed using ImageJ software (Version 1.54d, National Institutes of Health, Bethesda, MD, USA).

## 3. Results

### 3.1. MHY1025 Reduces LPS-Induced Oxidative Stress in NRK52E Cells

First, we evaluated the antioxidative effects of MHY1025 (Figure 1A). In the NRK52E renal tubular epithelial cells, there was no cytotoxicity detected under 10 μM of MHY1025 (Figure 1B). LPS treatment was used to evaluate the antioxidative effects of MHY1025 on the renal tubule epithelial cells. LPS treatment significantly increased the reactive oxygen species (ROS) in the NRK52E cells, and MHY1025 significantly reduced the ROS levels in these cells (Figure 1C). The ROS levels were visualized under a fluorescence microscope. MHY1025 pretreatment significantly reduced the LPS-induced ROS levels in NRK52E cells (Figure 1D). We measured cellular glutathione (GSH) levels. LPS treatment significantly reduced the GSH levels, whereas MHY1025 blocked this LPS-induced GSH decrease in the cells (Figure 1E). Collectively, these data indicated that MHY1025 significantly reduced LPS-induced oxidative stress in renal tubule cells.

### 3.2. MHY1025 Suppresses LPS-Induced NF-κB Activation and Chemokine Expression in NRK52E Cells

Oxidative stress is commonly linked to inflammation through the modulation of signaling pathways. NF-κB is a major transcription factor that regulates various inflammatory factors, including cytokines and chemokines. We further tested whether MHY1025 suppressed LPS-induced NF-κB activation (Figure 2A). MHY1025 pretreatment significantly reduced NF-κB activity as measured by luciferase activity (Figure 2B). We next detected NF-κB phosphorylation under the same conditions. LPS increased p65 phosphorylation in total cell lysates, whereas MHY1025 blocked LPS-induced p65 phosphorylation (Figure 1C). Nuclear lysates were separated to determine whether MHY1025 suppressed p65 translocation. LPS increased p65 translocation in the cells, whereas MHY1025 reduced p65 nuclear translocation (Figure 1D). Translocation of p65 was confirmed by immunofluorescence. MHY1025 pre-treatment effectively reduced p65 nuclear translocation (Figure 1E). Finally, we checked whether reduced NF-κB activation led to decreased chemokine expression in these cells. LPS significantly increased the gene expression of the major chemokines (*Cxcl1*, *Ccl2*, and *Il8*), whereas MHY1025 significantly reduced their expression (Figure 2F). These data indicate that MHY1025 suppresses LPS-induced NF-κB activation and chemokine expression in NRK52E cells.

### 3.3. MHY1025 Inhibits Fibroblast Activation Induced by TGFβ

Next, we evaluated the antifibrotic effect of MHY1025 on kidney fibroblasts. NRK49F fibroblasts were pretreated with MHY1025 and then treated with 10 ng/mL TGFβ1 (Figure 3A). TGFβ1 treatment significantly increased the gene expression of fibrosis-related proteins (*Acta2*, *Col1a2*, *Col3a1*, *Fn*, and *Vim*). MHY1025 treatment significantly blocked TGFβ1-induced fibrosis-related gene expression in the cells (Figure 3B). We further evaluated the protein expression of Collagen I and α-SMA. MHY1025 treatment significantly reduced Collagen I and α-SMA protein levels in NRK49F cells (Figure 3C). We also validated fibroblast activation using α-SMA immunofluorescence. TGFβ1-activated fibroblasts showed a significant increase in α-SMA expression, while MHY1025 reduced α-SMA expression in the cells (Figure 3D). These data show that MHY1025 has a direct antifibrotic effect on kidney fibroblasts.

### 3.4. MHY1025 Reduces Folic Acid-Induced Renal Damage and Tubule Dilation in Kidneys

Based on the antioxidative, anti-inflammatory, and antifibrotic characteristics of MHY1025, we further evaluated its efficacy using an in vivo kidney disease model. High-dose folic acid (FA) exposure led to kidney fibrosis characterized by excessive inflammation (Figure 4A). The FA groups showed an increased kidney weight/body weight ratio, whereas the MHY1025 group showed a smaller increase in this ratio (Figure 4B). FA treatment significantly increased the blood urea nitrogen levels, whereas the MHY1025 group showed a smaller increase in these levels (Figure 4C). We further evaluated changes in kidney damage-associated gene expression. FA treatment increased the gene expression levels of Hepatitis A virus cellular receptor 1 (*Havcr1)* and Lipocalin 2 (*Lcn2)*, and MHY1025 reduced their expression in the kidney (Figure 4D). Histological changes were further evaluated by H&E staining. The staining results showed that FA significantly increased tubule dilation in the kidney, whereas the MHY1025-treated group showed less tubule dilation (Figure 1E). Collectively, these data indicated that MHY1025 reduced FA-induced renal damage and tubule dilation.

### 3.5. Renal Inflammation Induced by FA Is Attenuated by MHY1025

The extent of renal inflammation was measured using the same animal model. The gene expression levels of pro-inflammatory chemokines (*Ccl2*, *Ccl3*, *Ccl5*, *Ccl7*, and *Cxcl1*) and cytokines (*Il1b*, *Il6*, and *Tnfa*) were significantly increased in the FA-treated kidneys, whereas the MHY1025-treated groups showed lower expression of these genes in the kidney (Figure 5A,B). The protein expression levels of p65 and phosphorylated-p65 were increased in the FA-treated group, and MHY1025 significantly reduced their protein expression in the kidney (Figure 5C). The protein markers of macrophages (CD68) and T cells (CD3) were also upregulated in the FA-treated group, indicating immune cell infiltration into the diseased kidneys (Figure 5C). The MHY1025 groups showed lower expression of these markers in the kidneys (Figure 5C). The expression of phosphorylated-p65 was detected histologically using immunohistochemistry (IHC). IHC analysis showed that the major cell type expressing p-p65 was tubular epithelial cells in the diseased kidney, and MHY1025 significantly reduced its expression in the kidney (Figure 5D). The extent of inflammation was confirmed by dual ISH methods. Some of the damaged tubules and interstitial cells showed highly increased *Ccl2* expression in the FA-treated kidneys (Figure 5E). Near the Ccl2-expressing cells, increased expression of macrophage marker (*Emr1*) was detected in the interstitial region between the tubules (Figure 5E). MHY1025 significantly reduced *Ccl2* and *Emr1* expression in the kidneys (Figure 5E). These data indicate that FA-induced renal inflammation is attenuated by MHY1025 in the kidneys.

### 3.6. MHY1025 Alleviates FA-Induced Renal Fibrosis in Kidneys

Finally, we examined whether the reduced inflammatory response led to the alleviation of renal fibrosis in the same animal model. The gene expression levels of fibrosis-related proteins (*Acta2*, *Col1a2*, *Col3a1*, *Fn*, *Tgfb1*, and *Vim*) were significantly increased in FA-treated kidneys, whereas the MHY1025-treated groups showed lower expression of these genes in the kidney (Figure 6A). The protein levels of COL1 and α-SMA showed a similar tendency with the gene expression changes in the kidney (Figure 6B). The extent of fibrosis was verified histologically. Sirius Red (SR) staining showed increased interstitial fibrosis in the FA-treated groups, and MHY1025 significantly reduced the SR-positive regions of the kidney (Figure 6C). IHC analysis also showed that α-SMA was highly expressed in the FA-treated kidney, and the MHY1025 group had less expression of α-SMA in the kidneys (Figure 6D). Finally, we detected expression of *Col1a1* and *Emr1* using dual ISH. FA-treated kidneys showed an increase in the expression of both genes in the same interstitial region, suggesting a relationship between inflammation and fibrosis (Figure 6E). The MHY1025-treated group showed a lower expression of these genes in the kidneys (Figure 6E). Collectively, these data indicated that, along with renal inflammation, renal fibrosis development was effectively reduced by MHY1025 in the kidney.

## 4. Discussion

The results of our study demonstrated the multifaceted beneficial effects of MHY1025 in kidney disease. Our investigation focused on elucidating the protective mechanisms of MHY1025 against various aspects of kidney injury ranging from oxidative stress and inflammation to fibrotic progression. The antioxidative properties of MHY1025 reduced the inflammatory response in the kidney epithelial cells. Furthermore, MHY1025 directly suppressed TGFβ-induced fibroblast activation and reduced fibrosis markers. Based on the in vitro observations, we further demonstrated the in vivo efficacy of MHY1025 in a folic acid-induced kidney fibrosis model. MHY1025 significantly reduced kidney inflammation and fibrosis Collectively, these data highlight the potential therapeutic value of MHY1025 in mitigating renal damage and dysfunction.

Oxidative stress is a fundamental pathological process implicated in various chronic diseases, including kidney disorders [19,20]. The kidneys are highly metabolically active organs, and their physiological functions, such as filtration, reabsorption, and excretion, involve complex cellular processes that generate ROS as byproducts [21]. Oxidative stress plays a pivotal role in the initiation and propagation of renal injury and contributes to inflammation, cellular damage, and fibrosis [22]. Kidneys are highly susceptible to various sources of oxidative stress, including ischemia/reperfusion injury, inflammation, metabolic disorders, and hypertension [23]. Oxidative stress triggers a cascade of detrimental effects on the kidney function and structure. ROS directly induce cellular damage by influencing cellular components such as lipids, proteins, and DNA, leading to cell dysfunction and death. ROS also contribute to the activation of pro-inflammatory pathways, promote fibroblast activation, and directly cause vascular endothelial dysfunction. Understanding the role of oxidative stress in kidney disease has prompted research on antioxidant therapies as potential interventions.

Inflammation, which is often associated with oxidative stress, is a hallmark of kidney injury [24]. During the initial phase, inflammation is activated as a protective response to eliminate the cause of the injury and facilitate tissue repair. However, during prolonged inflammation, fibroblasts can become overactive, leading to the excessive production of collagen and other extracellular matrix components. During extended inflammation, leukocytes originating from the bone marrow, particularly neutrophils and macrophages, play a central role in kidney inflammation [25]. Accumulation of these cells is a prominent hallmark of pro-inflammatory kidney disease. In addition, research has revealed a significant contribution of kidney cells activated within the kidney itself, including TECs, mesangial cells, podocytes, and endothelial cells. TECs play a pivotal role in initiating inflammatory responses during the progression of interstitial fibrosis [26]. Under injured or damaged conditions, TECs actively engage in pro-inflammatory reactions by producing chemokines. Based on these observations, the regulation of epithelial inflammation is an interesting target for modulating kidney inflammation and fibrosis. MHY1025 reduced oxidative stress and LPS-induced inflammatory responses in the renal tubular epithelial cells. MHY1025 significantly reduced NF-κB activation, translocation, and chemokine production in the epithelial cells. Since ROS play a significant role in NF-κB activation, it is reasonable to assume that the antioxidant properties of MHY1025 may have contributed to the control of epithelial inflammation.

In addition to its role in inflammation, oxidative stress has also been implicated in the development of kidney fibrosis. TGF-β is recognized as a pivotal facilitator of renal fibrosis, as it activates fibroblasts within the kidney. Therefore, it is an intriguing candidate for fibrosis treatment. While it is widely accepted that the Smad pathway plays a central role in TGF-β-induced fibrogenesis, recent findings suggest that oxidative stress exerts regulatory influence over TGF-β signaling through various pathways, including the Smad pathway [27]. Moreover, the interplay between TGF-β and oxidative stress creates a detrimental feedback loop. Consequently, pursuing interventions that address TGF-β-induced and ROS-driven cellular signaling emerges as an innovative strategy for tackling fibrotic conditions. We also found that MHY1025 exerts direct antifibrotic effects in TGF-β-induced fibroblast activation in NRK49F cells. MHY1025 effectively blocked TGF-β-induced collagen production and αSMA expression in fibroblasts.

Based on the in vitro observations, we further evaluated the MHY1025’s efficacy using an in vivo animal model of kidney fibrosis. The widely used folic acid-induced kidney fibrosis model is characterized by an excessive inflammatory response in epithelial cells. MHY1025 treatment significantly reduced renal damage, inflammation, and fibrosis. Nonetheless, our study was subject to several limitations. Our primary evaluation of the role of MHY1025 largely relied on cell lines, and it will be essential to utilize other in vitro models, including primary kidney cells, to validate its efficacy. Additionally, our investigation solely employed a single animal model of kidney fibrosis. To further substantiate the effectiveness of MHY1025, it is advisable to incorporate additional models such as the ischemia–reperfusion kidney injury model. Collectively, our results suggest that the administration of antioxidant compounds effectively prevented renal fibrosis in both in vitro and in vivo models of kidney fibrosis, indicating the therapeutic potential of antioxidants against CKD.

## 5. Conclusions

Collectively, our findings provide compelling evidence that the application of the thiobarbiturate-derived compound MHY1025 yields a significant and promising breakthrough in the mitigation of renal fibrosis. We observed these positive effects in a range of experimental settings, encompassing both in vitro and in vivo models of kidney fibrosis. These observations not only underscore the therapeutic potential of antioxidants but also offer a substantial stride towards combating CKD.

## Figures and Tables

**Figure 1 antioxidants-12-01947-f001:**
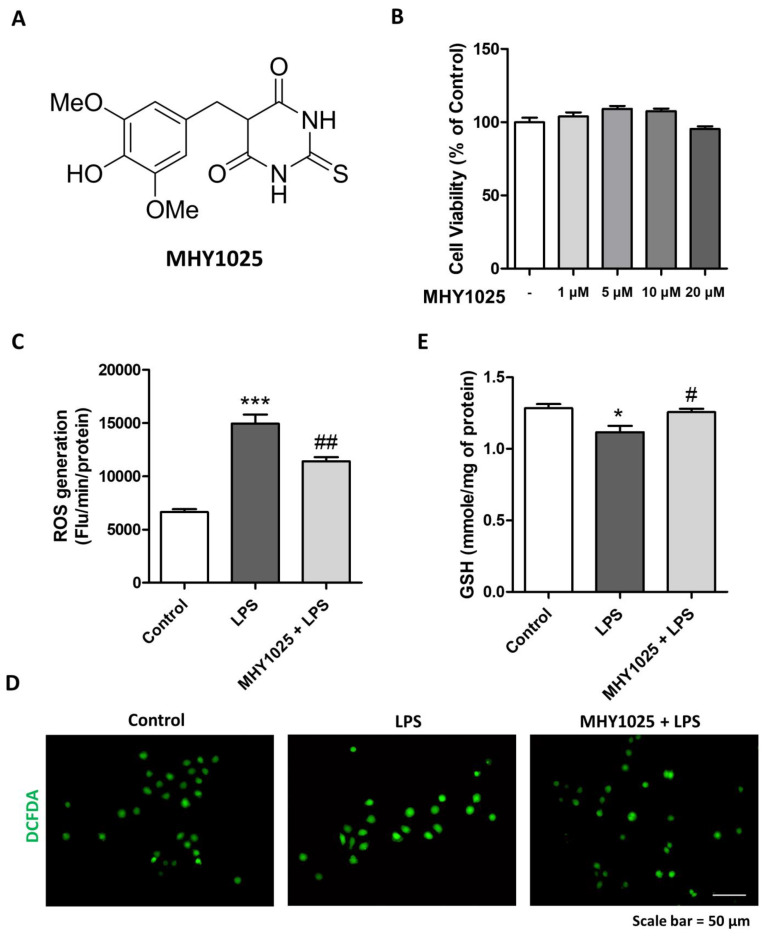
MHY1025 reduces LPS-induced oxidative stress in NRK52E cells: (**A**) MHY1025 structure; (**B**) Cytotoxicity was measured after treatment with different doses of MHY1025; (**C**) Oxidative stress was measured by DCFDA fluorescent dye in NRK52E cells treated with LPS and/or MHY1025. *** *p* < 0.001 vs. non-treated control group. ## *p* < 0.001 vs. LPS group; (**D**) Representative fluorescent image of DCFDA-stained NRK52E cells treated with LPS and/or MHY1025; (**E**) Cellular GSH levels were measured in NRK52E cells treated with LPS and/or MHY1025. * *p* < 0.05 vs. non-treated control group. # *p* < 0.05 vs. LPS group.

**Figure 2 antioxidants-12-01947-f002:**
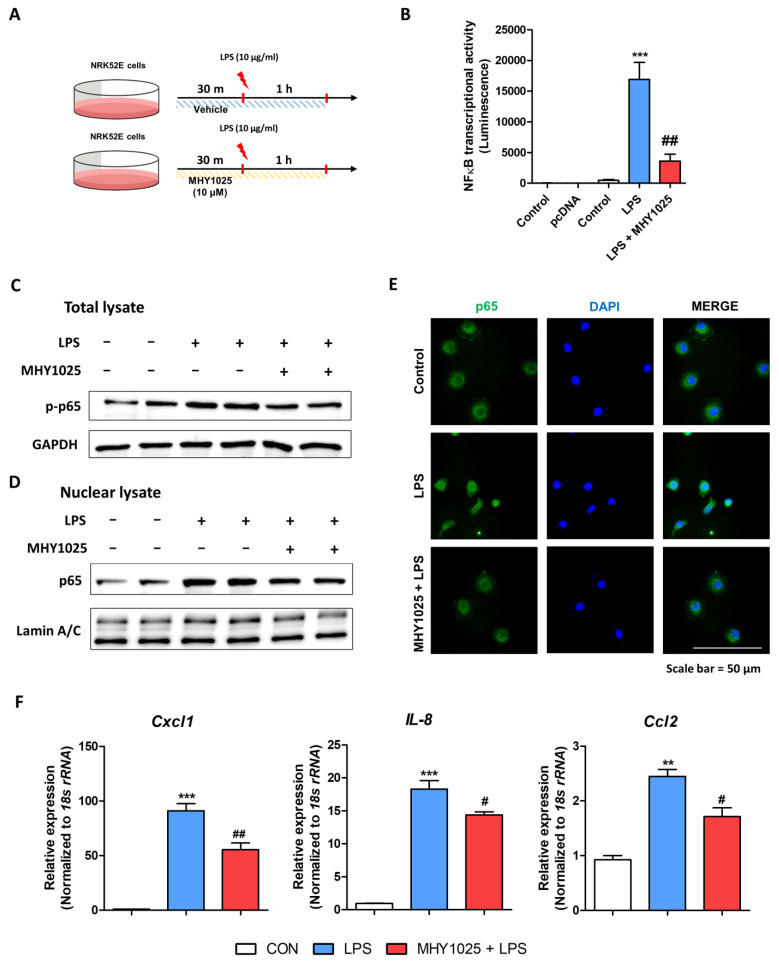
MHY1025 blocks LPS-induced NF-κB activation and chemokine expression in NRK52E cells: (**A**) Experimental scheme demonstrating the anti-inflammatory effects of MHY1025 in NRK52E cells treated with LPS; (**B**) NF-κB transcriptional activity was measured by luciferase assay under LPS treatment with or without MHY1025 treatment. *** *p* < 0.001 vs. non-treated control group. ## *p* < 0.01 vs. LPS-treated group; (**C**) Protein expression of phosphorylated p65 in total lysates of NRK52E cells under LPS treatment with or without MHY1025. GAPDH was used as the internal control; (**D**) Protein expression of p65 in nuclear lysates of NRK52E cells under LPS treatment with or without MHY1025. Lamin A/C was used as the internal control for nuclear lysates; (**E**) Representative immunofluorescence image showing the cellular location of p65 in LPS-treated NRK52E cells with or without MHY1025. The nuclei were counter-stained by DAPI; (**F**) Relative mRNA levels of inflammation-related chemokine genes (*Ccl2*, *Il8*, and *Cxcl1*) in LPS-treated NRK52E cells with or without MHY1025. ** *p* < 0.01, *** *p* < 0.001 vs. non-treated control group. # *p* < 0.05 ## *p* < 0.01 vs. LPS-treated group.

**Figure 3 antioxidants-12-01947-f003:**
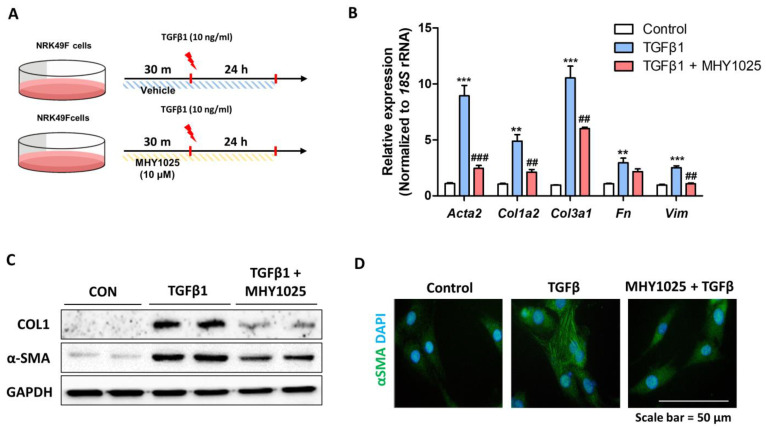
MHY1025 inhibits TGF-β-induced fibroblast activation in NRK49F fibroblasts: (**A**) Experimental scheme demonstrating the anti-fibrotic effects of MHY1025 in NRK49F cells treated with TGF-β; (**B**) Relative mRNA levels of kidney fibrosis-related genes (*Acta2*, *Col1a2*, *Col3a1*, *Fn*, and *Vim*) in TGF-β-treated NRK49F cells with or without MHY1025 treatment. ** *p* < 0.01, *** *p* < 0.001 vs. non-treated control group. ## *p* < 0.01, ### *p* < 0.001 vs. TGF-β-treated group; (**C**) The protein levels of αSMA and Collagen I were determined in TGF-β-treated NRK49F cells with or without MHY1025. GAPDH was used as a loading control; (**D**) Representative immunofluorescence images of α-SMA expression (green) in TGF-β-treated NRK49F cells with or without MHY1025. The nuclei were counter-stained by DAPI.

**Figure 4 antioxidants-12-01947-f004:**
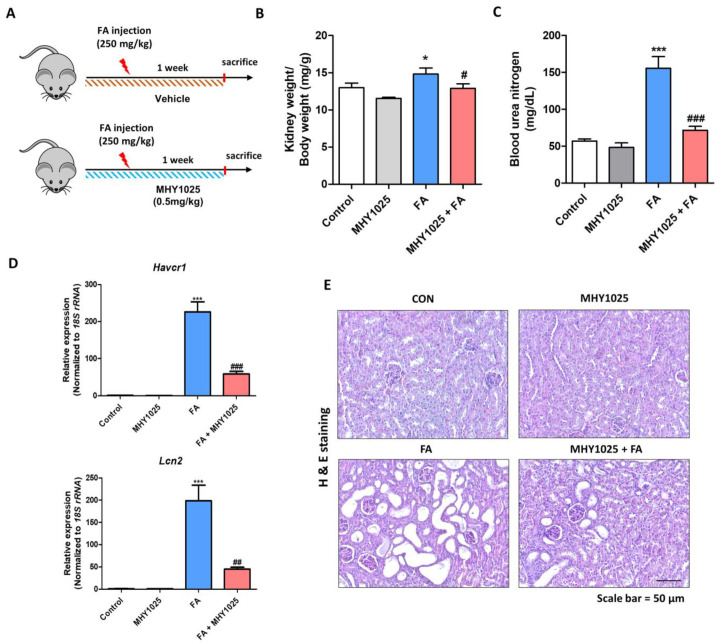
MHY1025 reduces renal damage in a folic acid (FA)-treated kidney fibrosis model: (**A**) Scheme for animal experiments; (**B**) Ratio of kidney weight to body weight at the end of the experimental schedule. * *p* < 0.05 vs. control group. # *p* < 0.05 vs. FA-treated group; (**C**) Blood urea nitrogen levels were measured to evaluate the kidney function of experimental animals. *** *p* < 0.001 vs. non-treated control group. ### *p* < 0.001 vs. FA-treated group; (**D**) Relative mRNA levels of kidney damage-related genes (*Havcr1* and *Lcn2*) in kidneys. *** *p* < 0.001 vs. non-treated control group. ## *p* < 0.01 vs. FA-treated group; (**E**) Representative H&E staining images of kidneys.

**Figure 5 antioxidants-12-01947-f005:**
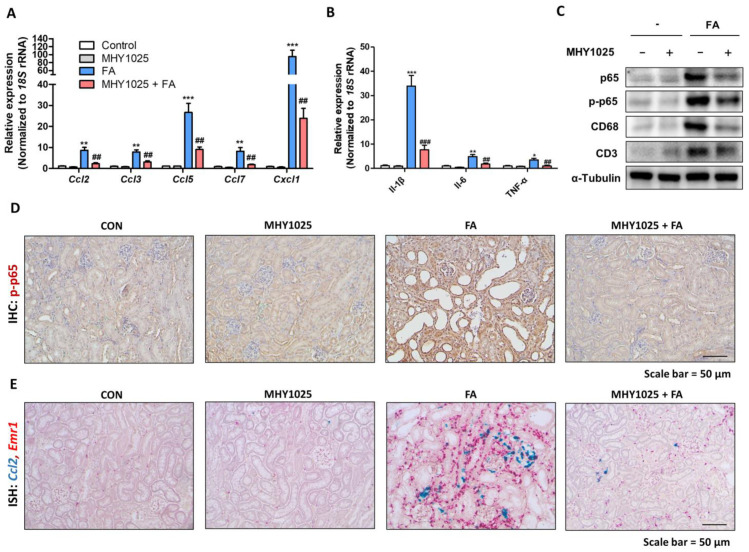
Renal inflammation induced by FA is attenuated by MHY1025: (**A**) Relative mRNA levels of inflammation-related chemokine genes (*Ccl2*, *Ccl3*, *Ccl5*, *Ccl7*, and *Cxcl1*) in FA-treated kidneys with or without MHY1025 treatment. ** *p* < 0.01, *** *p* < 0.001 vs. non-treated control group. ## *p* < 0.01 vs. FA-treated group; (**B**) Relative mRNA levels of inflammation-related cytokine genes (*Il1b*, *Il6*, and *Tnfa*) in FA-treated kidneys with or without MHY1025 treatment. * *p* < 0.05, ** *p* < 0.01, *** *p* < 0.001 vs. non-treated control group. ## *p* < 0.01, ### *p* < 0.001 vs. FA-treated group; (**C**) Protein levels of p65, p-p65, CD68, and CD3 were determined in FA-treated kidneys with or without MHY1025 treatment. α-tubulin was used as a loading control; (**D**) Representative IHC images of p-p65 expression in FA-treated kidneys with or without MHY1025 treatment; (**E**) Representative images of in situ hybridization with *Emr1* (red) and *Ccl2* (green) probe in FA-treated kidneys with or without MHY1025 treatment.

**Figure 6 antioxidants-12-01947-f006:**
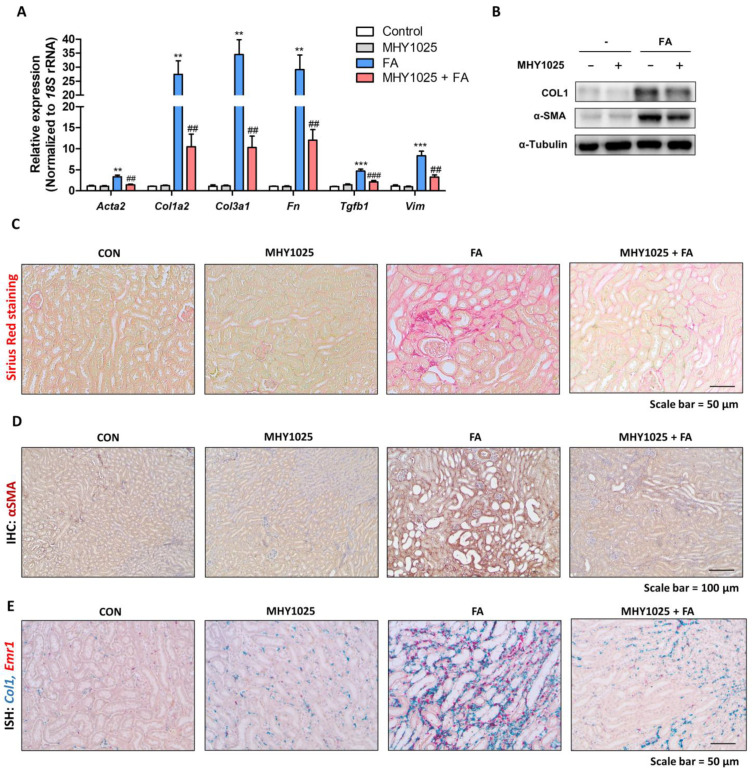
MHY1025 alleviates FA-induced renal fibrosis. (**A**) Relative mRNA levels of fibrosis-related genes (Acta2, Col1a2, Col3a1, Fn, Tgfb1, and Vim) in FA-treated kidneys with or without MHY1025 treatment. ** *p* < 0.01, *** *p* < 0.001 vs. control group. ## *p* < 0.01, ### *p* < 0.001 vs. FA-treated group. (**B**) Protein levels of Collagen Ⅰ and α-SMA were determined in FA-treated kidneys with or without MHY1025 treatment. α-tubulin was used as a loading control. (**C**) Representative images of Sirius Red staining in FA-treated kidneys with or without MHY1025 treatment. (**D**) Representative IHC images of αSMA expression in FA-treated kidneys with or without MHY1025 treatment. (**E**) Representative images of in situ hybridization with Emr1 (Red) and Col1a1 (Green) probe in FA-treated kidneys with or without MHY1025 treatment.

## Data Availability

Data is contained within the article and Appendix A.

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
