# Peer review of "Thiobarbiturate-Derived Compound MHY1025 Alleviates Renal Fibrosis by Modulating Oxidative Stress, Epithelial Inflammation, and Fibroblast Activation"

_antioxidants, 2023, doi:10.3390/antiox12111947_

Round 1

Reviewer 1 Report

Comments and Suggestions for Authors

This manuscript clearly demonstrated the beneficial/ameliorative effects of MHY1025, a thiobarbiturate derivative, in chronic kidney disease and fibrosis. There are a number of minor points that the authors need to address; however, as a whole the article is well written and well sourced, and the research approach is well described and sound.

Minor points:

Line 55: Define CTGF.

Line 76-80: Clarify what MHY1025 is in the context of thiobarbiturate derived compounds. Correct the spelling of thiobarbiturate in line 77. Also, add references for the "previous studies" that are referred to in the beginning and through the paragraph.

Figures 1, 2 and 3. Please indicate how many times (replicas) the studies were repeated. Please include a presentation of normalized densitometric data for the western blot analyses in Figs. 2 and 3. Please include whole pictures of your western blot images in the supplement. Line 299-300: Figure 4A is experimental scheme and has no information about fibrosis or inflammation.

Line 305: Please define Havcr1 and Lcn2.

Figures 4, 5 and 6. In schematic (Fig. 4A) please clarify that MHY1025 was administered at 0.5mg/kg/day for 5 days.  Please include the number of animals per experimental group (n=?). Also, indicate complet pictures of the western blots in Fig 5C and 6B in the supplementary results. Were these blots repeated using different preps and how many times? please include the number of animals and the number of replicas for westerns in the figure legends.

Major questions:

Although the effect of MHY1025 is quite dramatic in fibroblasts they are less so in epithelial cells. Could the less drastic effect in NRK52E cells be due to the concentration of MHY1025? Were higher concentrations of MHY1025(e.g., 20+µM) cytotoxic?

FA injury usually shows its peak injury and fibrosis by day 14. Why the investigators decide to stop the experiments on day 7 FA administration?

Going back to the protective effect of MHY1025 in fibroblasts vs epithelial cell lines and extending this to in vivo protective effect of this compound; the authors need to speculate/discuss the variable protective role imparted by this compound effect on reducing the response by interstitial and infiltrating cells vs that of TEC in the mediation and severity of CKD and fibrosis. It seems that although TEC response is important the inflammation and interstitial cell response is where the lion share of protection comes from. 

Author Response

Reviewer 1

This manuscript clearly demonstrated the beneficial/ameliorative effects of MHY1025, a thiobarbiturate derivative, in chronic kidney disease and fibrosis. There are a number of minor points that the authors need to address; however, as a whole the article is well written and well sourced, and the research approach is well described and sound.

Response to reviewer: Your insightful comments have been invaluable in enhancing the quality of the manuscripts. Thank you for your contribution.

Minor points:

Line 55: Define CTGF.

Response to reviewer: We defined CTGF.

Line 76-80: Clarify what MHY1025 is in the context of thiobarbiturate derived compounds. Correct the spelling of thiobarbiturate in line 77. Also, add references for the "previous studies" that are referred to in the beginning and through the paragraph.

Response to reviewer: Thanks for your comment. We added information about what MHY1025 is in the context of barbiturate derived compounds. We also corrected spelling of thiobarbiturate and added reference.

Figures 1, 2 and 3. Please indicate how many times (replicas) the studies were repeated. Please include a presentation of normalized densitometric data for the western blot analyses in Figs. 2 and 3. Please include whole pictures of your western blot images in the supplement. Line 299-300: Figure 4A is experimental scheme and has no information about fibrosis or inflammation.

Response to reviewer: Thanks for your comment. We added information about replicates in the material and method section. We also put whole pictures of western blot images in the supplement. Figure 4A shows overall scheme for the in vivo experiment schedule.

Line 305: Please define Havcr1 and Lcn2.

Response to reviewer: We defined Havcr1 and Lcn2.

Figures 4, 5 and 6. In schematic (Fig. 4A) please clarify that MHY1025 was administered at 0.5mg/kg/day for 5 days. Please include the number of animals per experimental group (n=?). Also, indicate complet pictures of the western blots in Fig 5C and 6B in the supplementary results. Were these blots repeated using different preps and how many times? please include the number of animals and the number of replicas for westerns in the figure legends.

Response to reviewer: Thanks for your comments. In the material and method section, we explained that MHY1025 was administered at 0.5mg/kg/day for a week. We added number of animals per experimental group in the material and methods section. We added whole blot of figure 5 and 6. The blots were repeated in every animal used in the experiment (n=7~8). We added number of animals in the material and method section. We added replicates number for western blots in the material and methods section.

.

Major questions:

Although the effect of MHY1025 is quite dramatic in fibroblasts they are less so in epithelial cells. Could the less drastic effect in NRK52E cells be due to the concentration of MHY1025? Were higher concentrations of MHY1025(e.g., 20+µM) cytotoxic?

Response to reviewer: Thanks for your insightful comment. We used same concentration of MHY1025 in both cell lines and we still do not have exact answer. The higher concentration over than 20 µM of MHY1025 had little cytotoxicity on the cells.

FA injury usually shows its peak injury and fibrosis by day 14. Why the investigators decide to stop the experiments on day 7 FA administration?

Response to reviewer: Thanks for your comment. We agree with the reviewer’s comment. Although the fibrosis extent peaks at day 14 in the individual mouse, we found that there was so much variation at day 14 (some mice were severely fibrotic, some mice showed regression, in the same group). We chose to see one week after FA injection in our setting.   

Going back to the protective effect of MHY1025 in fibroblasts vs epithelial cell lines and extending this to in vivo protective effect of this compound; the authors need to speculate/discuss the variable protective role imparted by this compound effect on reducing the response by interstitial and infiltrating cells vs that of TEC in the mediation and severity of CKD and fibrosis. It seems that although TEC response is important the inflammation and interstitial cell response is where the lion share of protection comes from.

Response to reviewer: Thanks for your insightful comment. We totally agree with the reviewer’s comment. Although our in vitro studies have shown the protective effects both in epithelial cells and fibroblasts, we still do not know how our molecular worked under the in vitro condition. FA treatment model directly damages epithelial cells, and damaged epithelial cells further recruits infiltrating cells, and activates fibroblasts. Based on FA model, we only can speculate that reduced oxidative damages in the epithelial cells showed protective effects under in vivo condition.

Reviewer 2 Report

Comments and Suggestions for Authors

I considered the manuscript entitled “MHY1025 alleviates renal fibrosis by modulating oxidative stress, epithelial inflammation, and fibroblast activation” by Jeongwon Kim, et al, that is intended to be published in Antioxidants journal.

I enjoyed the manuscript and the study, it is clearcut and simple. And I discovered anew promising compound with powerful properties. However, I do not see a real mechanism of action of MHY1025 in Introduction, that would support the hypothesis of the study. There is also not familial nature of this compound, apart from chemical structure. When seeking for more information in Discussion section, no further data were found either. Then, we must assume that MHY1025 is an acronym, which has a potent biological effect. Just an effect because a series of letters.. I believe that when someone is to be communicating to the scientific community, there must be total transparency.

Author Response

I considered the manuscript entitled “MHY1025 alleviates renal fibrosis by modulating oxidative stress, epithelial inflammation, and fibroblast activation” by Jeongwon Kim, et al, that is intended to be published in Antioxidants journal.

I enjoyed the manuscript and the study, it is clearcut and simple. And I discovered anew promising compound with powerful properties. However, I do not see a real mechanism of action of MHY1025 in Introduction, that would support the hypothesis of the study. There is also not familial nature of this compound, apart from chemical structure. When seeking for more information in Discussion section, no further data were found either. Then, we must assume that MHY1025 is an acronym, which has a potent biological effect. Just an effect because a series of letters.. I believe that when someone is to be communicating to the scientific community, there must be total transparency.

Response to reviewer: Thank you for your valuable feedback. We concur with the reviewer's remarks. In a prior publication authored by Moon et al., the research had already demonstrated the significant antioxidant properties of compounds with thio-barbiturate structures, particularly MHY1025, in liver cells. Building upon this foundation, our study aims to investigate the impact of MHY1025 on kidney cells and disease models. In the concluding section of the introduction, we underscore the previously documented attributes of MHY1025.

Reviewer 3 Report

Comments and Suggestions for Authors

This study investigated the MHY1025 role of anti-fibrosis in RTEC and mouse folic acid model. Some important information is missing from the readers.

First, what is MHY1025, although the authors provided its structure, still a lot of information should be provided. Did the authors synthesize it or purchase it? What are its targets? The author may write a paragraph in the Introduction for this information.

I feel this study tested everything, oxidative stress, inflammation, used kidney cells and fibroblasts. They don’t have a target to focus on. So again what are MHY1025’s targets, they should study its functions based on this.

The concentration of MHY1025 used to treat the cells should be provided in the Method.

Fig 1B, the concentrations are too close for the viability assay. Much lower and higher concentrations could be used. 

Author Response

This study investigated the MHY1025 role of anti-fibrosis in RTEC and mouse folic acid model. Some important information is missing from the readers.

Response to reviewer: We appreciate your valuable insights. We concur with the reviewer's comments, which have proven instrumental in enhancing the overall quality of the paper.

First, what is MHY1025, although the authors provided its structure, still a lot of information should be provided. Did the authors synthesize it or purchase it? What are its targets? The author may write a paragraph in the Introduction for this information.

Response to reviewer: Thank you for your comment. MHY1025 was synthesized using a thio-barbiturate structure as its foundation. The comprehensive synthesis rationale and methods are elaborated upon in a prior publication (Reference 20 by Moon et al.). In the concluding section of the introduction, we emphasize the established characteristics of MHY1025.

I feel this study tested everything, oxidative stress, inflammation, used kidney cells and fibroblasts. They don’t have a target to focus on. So again what are MHY1025’s targets, they should study its functions based on this.

Response to reviewer: Thank you for your comment. As previously mentioned, the primary focus of our study was on the antioxidant properties. A prior report by Moon et al. had already established the direct antioxidant properties of MHY1025. Given the link between oxidative stress and inflammation, we proceeded to investigate inflammatory responses in kidney cells, both in vivo and in vitro, as a complementary aspect of our research.

The concentration of MHY1025 used to treat the cells should be provided in the Method.

 Response to reviewer: Thank you for your comment. We added the concentration of MHY1025 used in the experiments.

Fig 1B, the concentrations are too close for the viability assay. Much lower and higher concentrations could be used. 

Response to reviewer: Thank you for your valuable feedback. We have incorporated cell cytotoxicity data for the 20 μM concentration of MHY1025. It's worth noting that we did not include data for concentrations under 1 μM of MHY1025, as these lower concentrations did not exhibit any cytotoxic effects.

Reviewer 4 Report

Comments and Suggestions for Authors

Experimental Design Clarity: The experimental design is quite clear and well-structured. However, there are some points that could be further clarified:

1. Ensure that appropriate control groups are included in the experiments. It's important to have a control group for each experimental condition to accurately assess the effects of MHY1025. For example, in Figure 1, you mention "vs. control group" and "vs. LPS group," but it's not clear what the control group represents. Specify whether it's untreated cells or cells treated with a vehicle control.

2. In the discussion section, address potential limitations of the study. For example, discuss any potential limitations related to the use of cell lines or animal models and how they may impact the translational relevance of the findings.

3. The authors may explore the in vivo metabolism pathways of MHY1025 and its impact on the hormonal system. This will help assess its stability and potential adverse effects during long-term use.

4. To gain a deeper understanding of the mechanism of action of MHY1025, the authors can conduct molecular biology experiments, such as protein-protein interactions and signaling pathway studies. This will help elucidate how MHY1025 mitigates oxidative stress, inhibits NF-kappaB activation, and reduces inflammation and fibrosis.

5. In the future, the authors can identify specific biomarkers associated with MHY1025 treatment response. The discovery of biomarkers can help predict patient responsiveness to MHY1025, allowing for personalized treatment strategies in clinical settings.

6. Compare MHY1025 with other antioxidants or anti-fibrotic agents in head-to-head experiments. This would help assess whether MHY1025 offers advantages over existing treatments and in what specific contexts it might be most beneficial.

Comments on the Quality of English Language

Minor editing of English language required

Author Response

Experimental Design Clarity: The experimental design is quite clear and well-structured. However, there are some points that could be further clarified:

Response to reviewer: We appreciate your valuable insights. We concur with the reviewer's comments, which have proven instrumental in enhancing the overall quality of the paper.

  1. Ensure that appropriate control groups are included in the experiments. It's important to have a control group for each experimental condition to accurately assess the effects of MHY1025. For example, in Figure 1, you mention "vs. control group" and "vs. LPS group," but it's not clear what the control group represents. Specify whether it's untreated cells or cells treated with a vehicle control.

Response to reviewer: Thank you for your comment. We have made the necessary modifications as per the reviewer's comments.

  1. In the discussion section, address potential limitations of the study. For example, discuss any potential limitations related to the use of cell lines or animal models and how they may impact the translational relevance of the findings.

Response to reviewer: Thank you for your comment. We added limitation of the study in the discussion section.

  1. The authors may explore the in vivo metabolism pathways of MHY1025 and its impact on the hormonal system. This will help assess its stability and potential adverse effects during long-term use.

Response to reviewer: Thank you for your comment. We wholeheartedly concur with the reviewer's feedback. Regrettably, due to constraints such as revision time limitations, we were unable to conduct more extensive experiments as suggested by the reviewer. Nevertheless, we acknowledge the significance of exploring the in vivo metabolic pathways of MHY1025, and we intend to address this aspect in our future research endeavors.

  1. To gain a deeper understanding of the mechanism of action of MHY1025, the authors can conduct molecular biology experiments, such as protein-protein interactions and signaling pathway studies. This will help elucidate how MHY1025 mitigates oxidative stress, inhibits NF-kappaB activation, and reduces inflammation and fibrosis.

Response to reviewers: Thank you for your comment. We totally agree with the reviewer’s comment. Again, due to constraints such as revision time limitations, we were unable to conduct more extensive experiments as suggested by the reviewer. We will try to look on protein-protein interactions and signaling pathway studies in the near future.

  1. In the future, the authors can identify specific biomarkers associated with MHY1025 treatment response. The discovery of biomarkers can help predict patient responsiveness to MHY1025, allowing for personalized treatment strategies in clinical settings.

Response to reviewers: Thank you for your suggestion. We also would like to identify specific biomarkers associated with MHY1025 treatment response. It will be helpful to make personalized treatment strategies in clinical settings.

  1. Compare MHY1025 with other antioxidants or anti-fibrotic agents in head-to-head experiments. This would help assess whether MHY1025 offers advantages over existing treatments and in what specific contexts it might be most beneficial.

Response to reviewers: Thank you for your comment. We agree with the comments that the efficacy of MHY1025 should be compared with other antioxidants or anti-fibrotic agents.

Round 2

Reviewer 2 Report

Comments and Suggestions for Authors

The only new information concerning the nature of MHY1025 is the following: Previous studies have shown that thio-barbiturate-derived compounds have antiox- 78 idant effects and prevent LPS-induced inflammation in the liver [20]. Thio-barbitureate- 79 derived compounds showed significant ROS- and ONOO-scavenging effects in test tubes 80 [20]. These compounds significantly reduced LPS-induced NF-κB activation in both mac- 81 rophage and liver injury models.

To me is not enough, again it appears just an acronym, not a compound

Author Response

The only new information concerning the nature of MHY1025 is the following: Previous studies have shown that thio-barbiturate-derived compounds have antiox- 78 idant effects and prevent LPS-induced inflammation in the liver [20]. Thio-barbitureate- 79 derived compounds showed significant ROS- and ONOO-scavenging effects in test tubes 80 [20]. These compounds significantly reduced LPS-induced NF-κB activation in both mac- 81 rophage and liver injury models.

To me is not enough, again it appears just an acronym, not a compound.

Response to reviewer: Thanks for your comment. In the figure 1, we shoed chemical structure of MHY1025. The word MHY comes from one of the co-corresponding author, “Hyung Ryoung Moon”. After synthesizing several compounds with thio-barbiturate structures, he named the compound after his name. We added some more explanation on the last paragraph of introduction. I hope this would give you the answer.

Reviewer 4 Report

Comments and Suggestions for Authors

I have no comments.

A good study.

Comments on the Quality of English Language

No comments

Author Response

Your insightful comments have been invaluable in enhancing the quality of the manuscripts. Thank you for your contribution.